# Recent Progress in Azopyridine-Containing Supramolecular Assembly: From Photoresponsive Liquid Crystals to Light-Driven Devices

**DOI:** 10.3390/molecules27133977

**Published:** 2022-06-21

**Authors:** Hao Ren, Peng Yang, Haifeng Yu

**Affiliations:** 1Key Laboratory of Applied Surface and Colloid Chemistry, Ministry of Education, School of Chemistry and Chemical Engineering, Shaanxi Normal University, Xi’an 710119, China; renhao@snnu.edu.cn; 2Institute of New Structural Materials, School of Material Science and Engineering, Peking University, Beijing 100871, China; 3Key Laboratory of Polymer Chemistry and Physics of Ministry of Education, Peking University, Beijing 100871, China

**Keywords:** azopyridine-based materials, supramolecular assembly, photoresponsive material, liquid crystal

## Abstract

Azobenzene derivatives have become one of the most famous photoresponsive chromophores in the past few decades for their reversible molecular switches upon the irradiation of actinic light. To meet the ever-increasing requirements for applications in materials science, biomedicine, and light-driven devices, it is usually necessary to adjust their photochemical property from the molecular level by changing the substituents on the benzene rings of azobenzene groups. Among the diverse azobenzene derivatives, azopyridine combines the photoresponsive feature of azobenzene groups and the supramolecular function of pyridyl moieties in one molecule. This unique feature provides pH-responsiveness and hydrogen/halogen/coordination binding sites in the same chromophore, paving a new way to prepare multi-functional responsive materials through non-covalent interactions and reversible chemical reactions. This review summarizes the photochemical and photophysical properties of azopyridine derivatives in supramolecular states (e.g., hydrogen/halogen bonding, coordination interactions, and quaternization reactions) and illustrates their applications from photoresponsive liquid crystals to light-driven devices. We hope this review can highlight azopyridine as one more versatile candidate molecule for designing novel photoresponsive materials towards light-driven applications.

## 1. Introduction

Light is absolutely charming, as it is an abundant and clean energy that has been widely utilized to manipulate photoresponsive materials remotely, precisely, and instantly. In the past few decades, the development of photoresponsive materials has attracted extensive attention in view of their wide-ranging applications, including nanotechnology, light-driven actuators [1,2], drug delivery systems [3], controlled biological systems [4,5], and many more [6,7,8,9]. In general, the photoresponsive materials are designed to acquire the reversibly/irreversibly changed molecular properties (such as polarity, size, and interior tension) in organic systems, causing the transformation of self-assembly structures and deformation of bulk materials. These photosensitive moieties in block copolymer systems and multiresponsive polymer systems have already been reviewed by Zhao [10] and Theato [11]. Among them, azobenzene derivatives are some of the most typical photoresponsive chromophores, existing in either the *trans* or *cis* conformations. The *trans*-form is the predominant conformation in dark conditions, for it is thermodynamically more stable than the *cis*-form. *Trans*-to-*cis* isomerization often occurs reversibly by applying a particular wavelength of light in their absorption bands, UV or blue light, respectively [12].

Typically, azobenzenes have been classified into three categories according to their substituent groups, which are classical “azobenzene” with no substituent groups, showing a relatively long lifetime of the *cis*-isomer [13]; “amino-azobenzene” with one electron-donating group, exhibiting a shorter *cis*-isomer lifetime; and “pseudo-stilbenes”, azobenzenes that possess an electron push–pull feature. This push–pull effect often leads to strong bathochromic shifting of the absorbance of chromophores and a shortening of its half-life of thermal reconversion to *trans* from *cis* [14].

Among all types of azobenzenes, one “good” chromophore for future applications should exhibit several characteristics. First, a rapid response is needed upon irradiation of actinic light with a specific wavelength so that the corresponding property change can be verified instantly [8,15]. Second, the reversible processes should be indispensable for most photo-actuated devices [16]. As mentioned above, the *cis*-to-*trans* thermal recovery of azobenzenes can only be tuned by introducing substituents on the aryls. The timescale of the lifetimes (τ) of *cis*-isomers vary from a few picoseconds to several days, depending on their substituents [13,17]. Third, the chromophore should have enough structural stability, as negligible decomposition occurs, even after prolonged light irradiation. Moreover, under high-intensity photoirradiation (>100 mW/cm^2^) [18], thermal energy is often produced from azobenzene-containing materials due to the so-called photothermal effect. Although the photothermal effect can be used in light-controlled devices [19,20], the chemical stability of chromophores should be emphasized.

Azobenzene-containing polymers in bulk and solution have already been systemically reviewed elsewhere [10,14,21,22]. The photoisomerization property of heteroaryl azo dyes was summarized recently by König and co-workers [23]. However, little attention has been paid to a class of a more versatile multiresponsive chromophores: azopyridine (AzPy). The structure of AzPy contains a pyridyl ring instead of a phenyl ring of azobenzenes (see Figure 1). Since the lone electron pair of the N atom in the pyridyl ring has no contribution of a π-electron conjugate, the N atom can act as an electron-withdrawing group of the conjugate system. In addition, the ionization of the pyridyl ring under acid conditions offers pH-responsive solubility and spectroscopic changes [24]. It is also worth mentioning that the N atom in the pyridyl ring is a powerful hydrogen/halogen bond acceptor and coordination ligand, a property employed extensively in the construction of supramolecular assemblies.

In this review, we emphasize the recent progress of AzPy-containing supramolecules with various interactions and versatile performances. We focus on the examples of current uses of AzPy as a building block in soft matter and materials science. We believe that the multifaceted AzPy chromophore can be an important and useful member of the photoresponsive family, enlarging the scope of multiresponsive materials.

## 2. Hydrogen-Bond Supramolecular Assembles

The lone-pair electron in AzPy enables it to interact with various hydrogen bond (H-bond) donors easily, acting as a building block for supramolecular assembly via H-bonding. The strength of a H-bond is around 2–160 kJ/mol based on its length and geometry [25]. The shorter the length and the closer the B-H···A angle to 180°, the stronger the H-bond, and vice versa [26,27]. The most widely investigated H-bond donors are carboxylic acid derivatives, and the equilibrium constant (K_a_) of the pyridyl/carboxylic acid complex was estimated to be five times stronger than that of the carboxylic acid dimer [28]. Because of their strength and directionality, H-bonds usually act as a powerful tool to guide supramolecular assembly in nature (e.g., proteins and DNA). In addition, a significant number of assemblies, such as liquid crystals (LCs), fibers, films, and gels, have been extensively studied. As shown below, we summarize the recent progress of AzPy-based supramolecules and the structures self-assembled via H-bonds.

### 2.1. Liquid Crystals

The mesogenic formation is the result of self-organization due to a proper combination of molecular shape and intermolecular function in a certain direction, which is generally regulated by molecular rigidness, the dipole–dipole interaction, and the steric hindrance effect as well as polar substituents [22]. A H-bond is much stronger than the dipole–dipole interaction, which provides an interestingly supramolecular interaction between the different components. The carboxylic acid and the pyridyl group have been generally used as H-bond donors and acceptors for the fabrication of supramolecular liquid crystals. The earliest example can be traced back to 1989 by Kato and Fréchet [29]. During the slow evaporation of the mixed solution of 4-butoxybenzoic acid and one four-substituent pyridine derivative, a new and extended mesogen was unexpectedly obtained through intermolecular H-bonding, with the molecular geometry directed along the long axis of the individual rod-like molecules. Other AzPy-carboxylic acid liquid–crystalline systems with small molecules were subsequently reported [30,31,32,33]. For example, Song and co-workers obtained a supramolecular liquid–crystalline complex from binary mixtures of 4-(alkoxyphenylazo) pyridines and 4-octyloxylbenzoic acid, where none of the pyridine-based derivatives were mesomorphic, but the H-bonded complexes were [34]. The structure–property relationship between the liquid–crystal parameters (e.g., the type of phase and the phase-transition temperature) was then systematically investigated by several groups [30,34,35].

In addition to the AzPy/carboxylic acid H-bond system, Pfletscher and Giese investigated the structural influence of phenol derivatives as H-bond donor moieties on the liquid–crystalline behavior of the AzPy-based supramolecular complex, as shown in Figure 2a [36]. After carefully analyzing the mesomorphic properties of 49 new H-bonded assemblies, they concluded that the linearly supramolecular architecture tended to form crystalline or smectic phases, while a V-shape structure had a high preorganization order of aromatics (blue) and aliphatic groups (red) bearing nematic phases. (Figure 2b) In their following works, hierarchical supramolecular liquid crystals were obtained by self-assembling different core units through H-bonding interactions. In these structures, AzPy derivatives act as H-bond acceptors, and aromatic polyols or polyphenols act as H-bond donors. For example, phloroglucinol [37], resveratrol [38], oxyresveratrol, butein, isoliquiritigenin, piceatannol [39], polycatenars [40], and other ortho-substituted phloroglucinols (e.g., 2-fluoro [41,42], cyan, and nitro [43]). The relationship between the core structure and the light-responsive liquid–crystal properties was systematically investigated. Combined with a detailed computational analysis with temperature-dependent FTIR results, they revealed an entropy-driven unfolding mechanism of the assembly [44]. Several H-bond liquid crystals exhibit rapid photoresponses and the broad-range blue phase, showing potential applications in organic optoelectronics as sensors or optical gates.

Very recently, several new types of AzPy-containing supramolecular liquid crystals with different geometries were reported, e.g., rod-like shape [33,45,46], chair/V-shape [47,48], and taper-shape [49] as well as bent-shape [49,50]. It was suggested that the chair-shaped conformers were more stable than the V-shaped isomeric complexes [47]. In addition, the rod-like conformation exhibited only an enantiotropic nematic phase over a broad range of temperatures, regardless of the terminal alkyl chain lengths at the pyridine-based component or the length of the flexible spacer on the benzoic acid derivatives (Figure 3) [33]. The molecular conformation and the thermal parameters of the complexes were also confirmed by theoretical calculations via density functional theory (DFT). Ahipa and co-workers reviewed the recent progress of heterodimeric H-bonded mesogens containing the pyridyl moiety [51].

In addition to the low-molecular-weight supramolecular system, researchers also took advantage of AzPy moieties in the light-controllable polymer to adjust the phase transitions of liquid crystals. For example, Zhao and co-workers developed an efficient strategy to fabricate photoactive liquid–crystalline materials through the self-assembly of AzPy side-chain polymers [52]. The AzPy-containing polymer (PAzPy) is totally amorphous upon thermal analysis. By contrast, when mixing with the aliphatic carboxylic acids, acetic (1COOH), hexanoic (5COOH) and decanoic acid (9COOH), the formation of the liquid–crystalline phase of the polymer was detected in all cases (Figure 4). Since then, polymer-based H-bond liquid crystals have been reported by many groups [53,54,55,56,57].

Yu and co-workers developed a supramolecular liquid–crystalline polymer microparticle (a diameter of 2~5 μm) with one AzPy-containing polymer (PM6AzPy) and a series of dicarboxylic acid (DA) compounds through H-bonding (Figure 5) [58]. The length of the diacid used can adjust the surface morphology of the obtained microparticles. Those diacids with long soft alkylene chains (9DA and 10DA) provide a wrinkled surface morphology for their relatively larger free volume during aggregation, while those diacids with shorter alkylene chains (1DA, 4DA, and 6DA) show smooth surfaces of the microparticles. The supramolecularly self-assembled microparticles showed mesogenic phases, and then the photoinduced liquid—crystal-to-isotropic phase transition was clearly observed upon UV irradiation, resulting in a deformed shape (Figure 5b).

### 2.2. Fibers and Gels

In 1999, Aoki and co-workers reported self-assembled fibers showing a uniform diameter of 200 nm derived from the pure compound of AzPy-containing carboxylic acid through the head-to-tail H-bonding in an alkaline solution [59]. The chemical structure of the assembly unit possesses both a carboxyl acid group as a H-bond donor and an AzPy moiety as a H-bond acceptor at each molecular terminus, as shown in Figure 6a. The intermolecular head-to-tail H-bonding and the inhibition of the π-π stacking of the substituent are the critical factors for the self-organization to form fibrous materials with high length-to-diameter ratios. The H-bonds between AzPy and carboxylic acid may offer the axial forces, while π-π stacking and dipole–dipole interactions provide the lateral intermolecular interaction forces (Figure 6b). Interestingly, UV irradiation of alkaline solutions of the AzPy-containing carboxylic acid (Figure 6a) resulted in a modification of the morphology from a fibrous structure to a needle-like structure, probably because the cis-isomer affected the nucleation process in the formation of the supramolecular fiber [60]. Then, AzPy derivatives with different spacer lengths, the substituent, and the solvent polarity were studied in detail, revealing that the morphological properties of these macroscopic self-assemblages can be controlled by the internal molecular structure and external stimuli, including heat, pH changes, light irradiation, and solvent polarity [61,62]. The weak interaction between the fiber formation as well as the possible molecular orientation is given in Figure 6b. These AzPy-based supramolecular fibers were then deployed as the template for fabricating metallic tubular materials where the inner diameter and the tubular morphology of the fibrous aggregates can be controlled by simply varying the amphoteric AzPy-containing carboxylic acids [63].

Different from mono (pyridyl) derivatives, supramolecular assemblies with bis(pyridyl) derivatives often result in relatively less mobile aggregations (e.g., fibers [64], gels [65], and crystalline materials [66]) for acting as the physical crosslinking points in aggregations. Naumov and co-workers reported an interesting example of a 4,4′-azopyridine-assembled crystalline material. The fibrillar crystal was obtained by a 2:1 co-crystal of probenecid and 4,4′-azopyridine (Figure 7), showing responses to multiple external stimuli, including heat, UV light, and mechanical pressure [66]. The crystal has excellent mechanical properties and can endure twisting, bending, and elastic deformation. Besides, the AzPy-containing crystal is capable of self-healing on heating and cooling upon UV irradiation because of the reversible *trans*–*cis* isomerization of the AzPy unit and the crystal-to-crystal phase transition (Figure 7c). This work presents a fantastic example of supramolecular co-crystals to overcome the main setbacks of molecular crystals, which may provide future applications as crystal actuators.

Recently, Zhao and co-workers successfully synthesized a chiral fiber by the co-assembly of phenylalanine-based enantiomers and an achiral AzPy through cooperative H-bonding interactions (Figure 8a) [64]. The obtained fibers have handedness inversion as well as controllable pitch and diameter. The helix inversion was achieved by the transition between the J-aggregation and the H-aggregation of bis(pyridyl) derivatives. Interestingly, the helical co-assemblies with opposite handedness could be obtained not only from the enantiomeric building blocks but also from the chiral monomers with the same configurational chirality by exchanging achiral additives, as shown in Figure 8c. Wei and coworkers designed a 2:1 co-crystal of (E)-4-((4-(propyloxy)phenyl)diazenyl)pyridine (APO3C) and tetrafluoroterephthalic acid (TFTA), and the resulting co-crystal exhibited photoinduced rotation under UV light. With the help of molecular simulation, the crystal habits and intermolecular interactions within these two assemblies were clarified. The pointer-like photomechanical rotation can be attributed to the opposing forces between the rotations of crystals caused by the *trans*–*cis* isomerization of APO3C molecules and the limited action of the diagonal arrangement [67].

On the aspect of the multiresponsive properties of AzPy-containing supramolecular assemblies, Feng and co-workers prepared a hydrogel co-assembled from a phenylalanine-based amphiphile (LPF2) and a bis(pyridyl) derivative (AP) through intermolecular H-bonds between amide/pyridine moieties and carbonyl groups [65]. The co-assembled hydrogel exhibited a macroscopic gel–sol transition in response to four distinct input stimuli: temperature, acid, base, and light (Figure 9). Based on these multiresponsive properties, a logic gate was created, which may promote the possibility of developing smart materials, such as gel-based tools.

Very recently, a supramolecular liquid–crystalline organogel was elegantly fabricated from an AzPy-containing polymer (PM11AzPy) and a large amount of oleic acid by Yu and co-workers through AzPy-carboxylic acid H-bonding (Figure 10a). Here, the oleic acid played the roles of both the solution and the gelator with H-bonded PM11AzPy. The PM11AzPy-based liquid–crystalline supramolecule exhibited a smectic C liquid–crystal phase from 38 to 58 °C (Figure 10b,c). In addition, the obtained organogel exhibited a multiresponsive gel−sol transition from external triggers, including temperature, UV light irradiation, and organic metal ions (Figure 10d). Based on such multiresponsive features, the organogel was successfully explored for recording holographic gratings, and the grating structures can be switched by manipulating the three external stimuli, indicating their potential applications as detectors or sensors [68].

### 2.3. H-Bonded Supramolecules in Solution

Aside from the delicate control of the phase and assembly structure via AzPy in the condensed state, AzPy chromophores can also be utilized in morphological regulation in solution. Han and co-workers reported the first example of an AzPy adjustable morphology transition from an amphiphilic block copolymer almost a decade ago [69]. In a water/tetrahydrofuran (H_2_O/THF) mixture solvent, amphiphilic poly (N-isopropylacrylamide)-*b*-poly{6-[4-(4-pyridyazo)phenoxy]hexylmethacrylate} (PNIPAM-*b*-PAzPy) self-assembled into giant micro-vesicles. Upon alternating irradiation of UV and visible light, reversible swelling and shrinking of the vesicles were clearly discovered under observation with an optical microscope. The maximum percentage increase in volume caused by UV light reached 17% (Figure 11). This change was explained by the reversible photoisomerization of the AzPy group inside the vesicle membrane, and the swelling degree could be adjusted by changing the light intensity. A similar photo-controllable transition behavior of vesical morphology was reported later by Lin and co-workers. The hydrophilic block copolymer composed of a hydrophilic poly (ethylene oxide) block and a hydrophobic polymethacrylate with photochromic AzPy moieties in the side groups was synthesized by atom transfer radical polymerization (ATRP) [70]. The copolymeric vesicles showed a photoinduced circular process including fusion, damage and defect formation, disruption, disintegration, and rearrangement in H_2_O/THF upon irradiation with UV light. Further studies on the same structure reflected that the *trans*–*cis* photoisomerization of AzPy in the vesicles was influenced by the water content in the mixture and the light intensity [71].

Yuan and co-workers synthesized one amphiphilic block copolymer containing N,N-dimethylaminoethyl and AzPy groups (PDMAEMA-*b*-PAzPy) that self-assembled into micelles with hydrophobic *trans*-AzPy cores and hydrophilic PDMAEMA shells, as shown in Figure 12 [72]. The micellar size exhibited triple tunable responses to temperature, pH, and light. Then, the light-responsive property of the polymer solution was investigated by fluorescence measurements of the micelles encapsulated with Nile red. The increase in fluorescence intensity after irradiation implies that the core of the micelle containing AzPy groups became more hydrophobic, which is totally different from the previous reports stating that the azobenzenes become more hydrophilic after UV irradiation [72]. The photochemical properties of AzPy chromophores in the copolymer micelle solution under different pHs were also investigated. In the acidic condition (pH = 2), the UV-Vis absorption spectrum hardly changed because of the quaternization of the AzPy moieties.

### 2.4. Photoisomerization and Relaxation of H-Bonded Supramolecules

As mentioned above, AzPy-containing H-bond supramolecular complexes have been extensively studied, from their aggregation morphology to their liquid–crystal properties. In this section, we will focus on the isomerization mechanism and its application of H-bonded supramolecules. It is well-known that the thermal relaxation of cis-azobenzene can be modulated by adjusting the electron delocalization via substituent groups at the benzene rings [12,13]. Specifically, electron-withdrawing groups with a push–pull effect would shorten the thermal relaxation time, while the electron-donating group prolongs it [13,73]. In AzPy, the lone electron pair of the N atom is not involved in the π-electron conjugate, which can act as an electron-withdrawing atom in the conjugate system, shortening the relaxation time. The delocalization of electrons intensively increases when the pyridine ring is in an ionized or quaternized state, resulting in an obvious decrease in the half-life of the cis-isomer (see detail in Section 5) [24]. In addition, the delocalization of the electrons also happens when the pyridine ring forms H-bonds with carboxylic acids or phenol derivatives. Recently, Gelbart and co-workers reported that the AzPy chromophore undergoes a very fast *cis*-to-*trans* thermal relaxation when the pyridine nitrogen is bound to a benzoic acid derivate. This fast recovery and decrease in half-life can be attributed to the enhanced push–pull effect of AzPy moieties under the supramolecular H-bonding state [15]. Such an exciting property can be brought into service for fabricating photo-driven devices with quick responsive deformation and fast recovery. For example, the AzPy-carboxylic acid H-bonded complex can produce supramolecular liquid–crystalline networks, which may act as a photoresponsive component in a liquid–crystalline polymer film containing diacrylate azobenzene monomer as the crosslinker. The films showed continuous wave motions upon photoirradiation, owing to the fast *cis*-to-*trans* thermal relaxation of the H-bonded AzPy complex. Thus, continuous, directional, macroscopic mechanical wave deformation can be achieved under constant light illumination. In addition, because of the photothermal effect of the liquid–crystalline film, the increased local temperature can also reduce the thermal relaxation time, promising applications in light-driven devices and self-cleaning surfaces [15].

In solution, the *cis*-to-*trans* relaxation of AzPy chromophores highly depends on the solvent property and the self-assembly structure. For example, Ren and co-workers recently demonstrated an exceptionally fast *cis*-to-*trans* thermal relaxation of the AzPy chromophore in an aqueous solution from a set of α-azopyridine-ω-dodecyl poly (N-isopropylacrylamide) homopolymers (C12-PN-AzPy) [24]. Figure 13 depicts the UV-Vis absorption spectra of the polymer under UV light irradiation in various conditions. The polymer self-assembles into flower-like micelles with closely packed n-dodecyl end groups as the core surrounded by hydrated PNIPAM chains as the shell. Upon UV irradiation of the solution (pH = 7), the *cis*-to-*trans* isomerization of the AzPy chromophore is greatly accelerated, with a half-life of *cis*-form of τ = 0.96 s, because of the formation of H-bonds between the AzPy moieties and N-H groups in the polymer micelles, where the existence of a strong push–pull effect by H-bonds greatly promotes the delocalization of the π electrons and accelerates the thermal relaxation of *cis*-AzPy. This fast relaxation of AzPy-terminated PNIPAM also helps to have a better understanding the micelle morphology [24]. In 2011, Tsuyoshi and co-workers observed the fast *cis*-to-*trans* relaxation in an AzPy-containing PNIPAM copolymer system in water but unfortunately overlooked it [74]. Changing the pH value of a solution will also change the state of the pyridine ring, which is in protonated state at pH = 3 with a very fast half-life of the *cis*-form (τ = 2.3 ms). By contrast, when the solution is modulated to pH = 10, the *cis*-to-*trans* relaxation of AzPy takes over 2 h, indicating that the AzPy groups are in a free state since the additional hydroxyl anions may replace the H-bond between AzPy and the amide hydrogen of the PNIPAM repeat unit. Such a H-bond-caused fast relaxation in solution was recently reviewed [75].

Benefitting from the controllable ionization and H-bonding formation/deformation in micelles, the *cis*-to-*trans* recovery rate can be easily modulated from hrs to ms by changing the solution pH. This pH- and photoresponsive chromophore was pressed into service as the modulator of multiresponsive polymers. A typical example is to actuate the solution property of PNIPAM, which undergoes a reversible phase transition upon heating over ∼32 °C, corresponding to the coil-to-globule collapse of the polymer chains. Thus, the macroscopic transition temperature is regarded as the cloud point (T_cp_) [76,77,78]. For example, an AzPy-terminated PNIPAM undergoes temperature-, pH-, and UV-light-driven phase transitions in aqueous media, as shown in Figure 14 [79]. When AzPy terminal groups are in the H-bonded state in polymer micelles under neutral solutions, the turbid polymer solutions at 20 °C (above the T_cp_ with the *trans*-form AzPy end group) become transparent upon a short irradiation of UV light since light converts *trans*-AzPy into a highly polar *cis*-AzPy isomer. In the dark, solutions of pH 7 become turbid again within a few seconds, as shown in Figure 14 (pH = 7). The above process can be repeated many times without obvious fatigue, showing good reversible characteristics. In addition, UV light irradiation does not influence the turbidity of the solutions at pH = 3 since the *trans* and *cis* isomers undergo an extremely fast exchange. While in solution (pH = 10), the photoinduced increase in the cloud point does not exhibit a quick recovery because of the long-lived *cis*-AzPy moieties.

In addition to experimental achievements, theoretical calculations have been regarded as a powerful tool for the design of liquid crystals and supramolecular materials. For example, Jimmy and coworkers recently predicted the dynamic photoreaction pathways of azobenzene with the help of first-principles simulations of nonadiabatic dynamics following excitation to both the π-π* and the n-π* states. They indicated two distinct S_1_ decay pathways, the reactive twisting and the unreactive planar pathway. Moreover, the unreactive pathway upon π-π* excitation largely accounts for the wavelength-dependent behavior of azobenzene [80]. Karolina and coworkers utilized DFT to optimize the ground and transition state geometries, the densities of states, and the electronic structures of azobenzene structures, indicating that the possible mechanism of the cis–trans isomerization process for 4-(4-hydroxyphenylazo)pyridine has a different relaxation time than other azo-compounds [13].

## 3. Halogen-Bond Supramolecular Assembles

Similar to the H-bond, the AzPy group, as an electron-rich Lewis base, can act as a halogen-bond (X-bond) acceptor due to the lone pair of electrons of a nitrogen atom in the pyridine ring. The nitrogen atom interacts non-covalently with an electropositive Lewis acid (H-bond donor), usually a σ-hole of a halogen atom, to form an X-bond [81,82]. The X-bond strength ranges from 10 kJ/mol to 150 kJ/mol, depending on the interaction partners, which is slightly weaker than the corresponding H-bond and can be modulated by the composition of the donor atoms as well as its substitution groups. Typically, X-bond strength has a general order of I > Br > Cl > F, where a fluorine atom can only be an X-bond donor when attached to strong electron-withdrawing groups [83]. In addition, X-bonding is more directional than H-bonding because the σ-hole of a halogen atom is narrowly confined on the elongation of the R-X covalent bond axis. Hence, the B-X···A angle is closer to 180° than a H-bond [84,85]. The first report of the X-bond can be traced back to 1863 when Guthrie described the formation of the NH_3_∙∙∙I_2_ complex [86]. After that, the bond energy and geometry details of X-bonds were investigated from the solid to gas phases, which was reviewed by Legon [87]. In 2004, Bruce and co-workers reported an X-bonded liquid–crystalline complex by the equimolar mixing of the non-mesomorphic components of 4-alkoxystilbazole with pentafluoroiodobenzene [88]. Then, the following research was reviewed by Giuseppe and co-workers [81]. Very recently, X-bonding has been successfully applied in designing novel photoresponsive liquid crystals from small molecules to polymers. Benefiting from the strength and more directional nature of X-bonds, supramolecular chemists can easily design and prepare diverse smart materials. This part will focus on the X-bond supramolecular assemblies based on AzPy derivatives.

In 2014, Chen and co-workers reported the first example of AzPy-containing X-bonded liquid crystals, where halogen molecules (I_2_ or Br_2_) act as X-bond donors and pyridine in the AzPy-containing molecule with different alkyl chains functions as the X-bond acceptor, as shown in Figure 15 [89]. In addition, the bromine-bonded liquid crystals were obtained with a high mesophase stability, indicating that the N-Br interaction is strong enough to form ordered mesophases in the specific system. Upon UV irradiation, iodine-bonded liquid crystals show a reversible photoinduced phase transition but are not detected in bromine-bonded liquid crystals. These photochemical processes are completely reversible, and the iodine-bonded liquid crystals appeared when the visible light was exposed on the isotropic samples.

X-bonds between AzPy and fluorine substitute chemicals are another important category. For example, Alaasar and co-workers designed a photoswitchable liquid–crystalline aggregate with an X-bonding formation between a non-mesogenic tetrafluoroiodobenzene as an X-bond donor and non-mesogenic AzPy derivatives as X-bond acceptors [90]. Interestingly, these X-bonded polycatenars exhibit enantiotropic liquid–crystal phases over a wide range of temperatures, which are the widest among all photoresponsive perfluoroaryliodide-based supramolecularly X-bonded liquid crystals. A similar structure with photoresponsive iodine-bonded liquid crystals based on AzPy derivatives with a low phase-transition temperature was also reported recently by Du and co-workers [91].

Changing the degree of fluorination at the X-bond donor of the supramolecular liquid crystal allows for the fine-tuning of the X-bond strength and thereby provides control over the mesophase temperature range. In 2018, Saccone and co-workers evaluated a series of X-bond donors of the fluorine substitute iodobenzene (D1-7 in Figure 16) [92]. With the help of a DFT calculation, the molecular electrostatic potentials of the iodine atom increase (more positive) as the number of fluorine atoms increases, demonstrating a simplistic “the shorter, the stronger” view of the H-bonding interaction. However, at least three fluorine atoms must be present to ensure the efficient polarization of the X-bond donor and the mesophase formation. This work offers a method for fine-tuning the X-bond strength by changing the degree of fluorination at the X-bond donor. As suggested, it is better to use donors and acceptors with long-lived metastable states and liquid–crystal phases that occur close to room temperature with low clearing points for stabilizing the photoinduced isotropic state of X-bond complexes. 

Hu and co-workers recently reported an AzPy-containing liquid–crystalline gel through X-bonding between an AzPy-C10 and 1,4-tetrafluorodiiodobenzene (TFDIB), as shown in Figure 17a [93]. Upon irradiation with UV light, the gel shows a gel–sol transition as well as a morphology change from flake to peony-like due to the light-induced *trans*–*cis* isomerization of the AzPy moiety. This remarkable photo-modulated morphology transition can be attributed to the variations in the *cis*-isomer content and X-bond strength. Using the same construction strategies, Tong and co-workers introduced a visible light-responsive gel structure via X-bonding by the mixing of AzPy-Cn (*n* = 8, 10, and 12) as an X-bond acceptor and 1,2-bis(2,3,5,6-tetrafluoro-4-iodophenyl) diazene (BTFIPD) as both the X-bond donor and visible-light-responsive moiety (Figure 17b) [94]. The obtained gel has a visible-light-responsive gel-to-sol transition under green light irradiation because the BTFIPD moieties contain an electron-withdrawing group (fluorine) to azobenzene at the ortho-position. This method provides a useful strategy for the preparation of a visible-light-triggered phase transition of supramolecular materials, in particular application scenarios, from energy conversion to information storage.

Very recently, Chen and co-workers successfully prepared an X-bond-based poly(ethylene oxide) and AzPy-containing block copolymer (PEO-*b*-PAzPy) by co-assembly with 1,2-diiodotetrafluorobenzene (1,2-DITFB), as shown in Figure 18a [95]. Macroscopically ordered nanocylinders in a hexagonal packing were homogeneously dispersed in a polymer film displaying a smectic-A (SmA) phase due to the X-bond self-organization. Interestingly, the efficient photoalignment and photo-reorientation of the nanocylinder array in the supramolecular X-bond liquid–crystal block copolymer film were successfully obtained by manipulating linearly polarized light irradiation, as shown in Figure 18b. The robust nature of mesogens in the X-bonded state and the enhanced directional ordering of the *cis*-isomer can be a reason for this photoalignment. It is an extremely rare example of the elegant maneuverability of the nanostructures of polymers by the emerging supramolecular interaction.

## 4. Coordination Interaction with Metal Ions

As a neutral electron-donor ligand, pyridine and its derivatives easily form a coordination bond with the vacant orbital of various metals. Thus, pyridine derivatives are important ligands in coordination chemistry that have been employed with all transition metals in producing pyridine–metal complexes [96,97]. The coordination complex with AzPy combines the physical characteristics of metal ions with the well-known properties of AzPy ligands, such as photoinduced isomerization, the self-organization of mesogens, and the formation of metal–organic frameworks (MOFs). The synthesis, characterization, and application of AzPy (usually 4,4′-azopyridine) MOFs were recently investigated and reviewed [98,99,100]. These complexes also show potential applications in the areas of colorimetric detection, water harvesting, and CO_2_ capture [99,101,102]. In this section, we will focus on the photo-switchable metallo-mesogens, especially their photoresponsive properties and the mesogenic structures of the supramolecular complexes.

In 2002, Das’s group synthesized a series of AzPy-containing silver complexes exhibiting nematic, smectic, and cholesteric liquid–crystalline phases, respectively [103]. The complexes were prepared by the stoichiometric reaction of silver dodecyl sulfate and the corresponding ligand in anhydrous methanol or dichloromethane at an ambient temperature in darkness. The isotropization temperatures have an odd–even effect against the alkane chain lengths of AzPy derivatives. The silver-based mesogens are relatively stable under UV irradiation, and the lifetime of *cis*-AzPy in silver-based liquid–crystal complexes is less than that in the non-coordination state [103]. In the following work of the same group, hexagonal columnar liquid–crystalline phases were detected from AzPy-containing hexacatenar silver complexes [104]. The photoisomerization study of these complexes indicates that the rate of the *cis*-to-*trans* thermal relaxation of silver complexes was ~50–100 times faster than that of AzPy-containing compounds, which can be traced back to the increase in donor–acceptor electron delocalization in AzPy-Ag complexes.

Cui and co-workers synthesized diblock copolymers (PS-*b*-PAzPy) containing polystyrene (PS) and a polymethacrylate with an AzPy side group [105]. The amorphous photoactive block of PAzPy can be transformed into a liquid–crystalline phase through self-assembly with different carboxylic acids through H-bonding. Moreover, with the coordination interaction of zinc-tetraphenylporphyrin (ZnTPP) between the metal ions and the pyridyl group of the AzPy side-chain polymer, the complex becomes both photoactive and electroactive, as shown in Figure 19. In addition, AzPy-containing polymers efficiently increase the glass transition temperature (T_g_) and thermal stability when coordinating with metalloporphyrins (ZnTPP or CoTPP). Then, Dahmane and co-workers reported a strategy of optically and electrochemically active supramolecular polymers through a coordination interaction [106]. The *trans*-to-*cis* photoisomerization of the AzPy chromophore was restricted after a coordination interaction in the solid state. While an organic solvent (a mixture of dichloromethane and acetonitrile, 1/5, *v*/*v*) showed different behaviors, the reversible *trans*-to-*cis* photoisomerization of AzPy upon light irradiation was found to alter the equilibrium of the axial coordination between pyridine groups and ZnTPP in solution, resulting in photo-regulable redox potentials and fluorescence emission of the metalloporphyrin [107]. Very recently, another AzPy-containing block copolymer was reported to inhibit the photoisomerization of AzPy moieties when it formed a coordination interaction with a four-coordinate cobalt (II) Schiff base in toluene [108].

In addition, Zhao and colleagues fabricated cholesterol-AzPy conjugate organogels by H-bond-based self-assembly [109]. The obtained organogels have a photo-controllable dimensional transition from 2D microbelts to 1D nanotubes and finally to 0D nanoparticles, as shown in Figure 20a. The driving force for this dimensional transformation is the photoisomerization of the 4-AzPy unit. Interestingly, with the addition of metal ions into the organogels, the self-assembled laminar aggregates were noticed to have a helicity inversion based on the coordination between the metal ions and the pyridyl unit. Specifically, Ni ^2+^, Mg^2+^, and Eu^3+^ ions turned the gel into left-handed twisted nanoribbons, while right-handed nanostructures were obtained from the Cu^2+^- and Bi^3+^-based metallogels (Figure 20b).

Very recently, Peng and co-workers successfully introduced an AzPy-Zn(II) coordination system into polydimethylsiloxane (PDMS) to obtain a self-healing polymer that demonstrates excellent light-healing properties, even after three cutting/healing cycles at a mild temperature (40 °C) in various harsh conditions (e.g., underwater and subzero temperature), as shown in Figure 21 [110]. After cutting off the elastic complex, breakage can be renewed in 24 h because of the coordination bond and the photoisomerization of AzPy moieties. The repairing efficiency was around 93.4% after irradiation by 365 nm and 450 nm light on the cut surface. In addition, the obtained supramolecular elastomer with outstanding sensitivity to strain makes it an excellent candidate for light and stress-simulated flexible devices.

Another recent example of AzPy-containing coordination gel was reported by Li and co-workers. They employed C3-symmetric AzPy ligands and Ag(I) to form a coordination polymer [111]. Due to the presence of dynamic coordinating bonds, the obtained polymer exhibited self-healing and multi-stimuli responses to heating, light, mechanical shearing, and chemicals. Guo and co-workers prepared a molecular crystal from an AzPy small molecular 4-(4-(6-Hydroxyhexyloxy) phenylazo) pyridine (6cAzPy) and Zinc (II). The obtained crystals exhibited a fascinating photomechanical bending motion, which was related to the *trans*-to-*cis* photoisomerization of the AzPy derivatives in the crystalline phase. After forming the metal-AzPy complex, this motion was enhanced because of the looser packing of the molecules inside the complex crystal [112].

## 5. Quaternization Reaction

As a Lewis base with a pKa at ~4.53 [113], the AzPy chromophore is able to react with a variety of organic/inorganic acids. Besides, the pyridine group in AzPy, moieties are capable of reactions with alkyl halides, resulting in a pyridinium salt through a quaternization reaction. Undoubtedly, the physicochemical properties (e.g., electron delocalization, water-solubility, the photoisomerization mechanism, and the hydrogen/halogen/coordination condition) can be regulated by simply changing the solution pH into an acidic environment or reacting with alkyl halides to form azopyridinium salts.

In 2010, Garcia-Amoro and co-workers reported a modified mechanism for the photoisomerization of quaternized AzPy chromophores by analyzing the photochemistry property of a set of azopyridinium methyl iodide salts [114], as shown in Figure 22. After quaternization, the absorption band had an obvious redshift of 50–60 nm from 350 nm in the native state to ~ 410 nm in the quaternized state. This notable redshift is related to a strong charge transfer from the alkoxy group to the positively charged nitrogen atom due to the push–pull effect. Interestingly, the *cis*-to-*trans* thermal recovery of azopyridium was obtained within 130–450 ms, much faster than that reported previously for other push–pull azobenzene-doped nematic mixtures. The lifetime of *cis*-azopyridium is too short to be detected by a normal UV-Vis spectrometer, and a transient absorption spectrum was recommended (Figure 22b). Figure 22c shows the proposed mechanism of the fast thermal relaxation, where the partial breaking of the N=N bond of the azobenzene moiety occurred due to an electron transfer from the alkoxy group to the positive charge center. Hence, the partial break of N=N bonds facilitates the rotation around itself to recover the more stable *trans* conformation [114]. The substituent group effect of this fast relaxation was investigated and reviewed by the following work in the same group [73,115,116]. Since this fast isomerization is able to be conducted at room temperature, which is very similar to the physiological conditions, the possibility of the design of the AzPy chromophore in biological and medical applications and in opto-electronic switches was also explored [117,118]. Very recently, this fast recovery of protonated AzPy in polymer aqueous solution was also investigated [24,79]. The self-assembly structures and the solution cloud point of AzPy-terminated PNIPAM homopolymers can be easily controlled by verifying the pH value of the solution.

On the aspect of the acid–base reaction, the azopyridium-based nanofibers were first reported by Zhou and co-workers [119,120] with a series of low-molecular-weight amphiphilic azopyridiniums and dodecylbenzenesulfonic acid or its salt (Figure 23). The film of the fabricated hybrid fibers shows interesting electrical conductivity in the order of 1.0 × 10^−7^ to 1.0 × 10^−5^ S/cm, which was attributed to the existence of freely movable ions [119]. Chen and co-workers fabricated a series of AzPy-based nanofibers with inorganic acids in various organic solvents. The fiber morphologies can be controlled by the acid dissociation constant (pKa values) [121].

Very recently, Xue and co-workers reported the CO_2_ response property of AzPy moieties when introducing CO_2_ into an AzPy-containing (9~13% mol content) copolymer P(NIPAM-*co*-PAZO-*co*-EGMA) [122]. The cloud point of the polymer solution (2 mg/mL) increased from 49 °C to 62 °C after injecting CO_2_ gas due to an enhancement of the water solubility of the protonated AzPy component in an acidic condition caused by carbonate formation. The reversible transparency was recorded from the AzPy-containing copolymer solution at 60 °C with alternating injections of CO_2_ gas and inert Ar_2_ gas into the solution. On the aspect of the photochemistry of AzPy in the copolymer solution, there were nearly no absorbance changes upon UV light irradiation under acidic conditions. In the the neutral condition, it showed a slight decrease in the absorbance peak at around 310 nm.

Besides bringing the pH-sensitive characteristics into the supramolecular system, the quaternization reaction of AzPy was also used as a crosslinking site. For example, Li and co-workers designed photo-activated bimorph composites with AzPy side-chain liquid–crystalline polymers and Kapton film (commercially available polyimide) to mimic the circadian rhythm behavior of Albizia julibrissin leaves, as shown in Figure 24 [123]. After adding a certain amount of 1,4-diidonetetra-butane (DIB) as the crosslinking agent, the AzPy group gradually quaternized with DIB. Interestingly, the photoresponsive rate of the crosslinked copolymer was increased by changing the AzPy group into the famous type of push–pull azobenzene.

## 6. Conclusions and Outlook

In conclusion, this paper reviewed the AzPy-containing supramolecular materials based on the hydrogen/halogen bonds and coordination bonds as well as ionic bonds. A great many outstanding examples of each interaction were reviewed, from delicate synthetic strategies to the structure–function relationship. These supramolecular assembles expand the synthesis strategies for multiple responsive materials, including responsive liquid–crystalline compounds, films, fibers, and gels in the condensed state, the micelles in the solution state, and even light-driven actuators based on composite materials. Some breakthroughs have been achieved in the fabrication of photoinduced oscillation films as self-cleaning surfaces through H-bonding [15], rearrangements of nanocylinder arrays through X-bonding [95], and self-healing of the elastic complex through the coordination interaction [110].

Although significant progress has been achieved in this area in the past decades, the following issues still need to be addressed to promote the development of AzPy-containing materials from theoretical study to practical applications. First, increased efforts should be directed at regulating wide time-scale relaxation and precise space scales to match different application scenarios by carefully controlling the ligands. For example, the bond strengths and the operating wavelengths of AzPy-containing supramolecular complexes can be controlled by the structures of the donor and acceptor species [36,92]. The thermal recovery of *cis*-isomerism can also modulate from ms to hrs by controlling the electron delocalization states of light-responsive moieties in some specific examples [24]. In addition, AzPy-containing materials derived by visible light may extend their applications in the fields of medical and life sciences. Borchers and co-workers gave an excellent example of a visible light photocarving of X-bonded co-crystals with micrometer-scale precision [124]. Second, it is of great significance to fabricate AzPy materials showing high mechanical/light stability. Currently, most AzPy supramolecular aggregates are still limited to achieving high mechanical strength, which can only be utilized by compositing with high-strength polymers, such as polyimide, polyethylene (PE), and other films. However, the stability and durability of its composition materials have a restriction in its application, especially in wearable devices. Konieczkowska and co-workers have made some explorations in this regard by introducing AzPy into the polymer main chain or crosslinker to form an elastomer [125,126,127]. Finally, based on the improved mechanical properties and precise modulations of AzPy materials, they may have extensive application prospects in the field of 4D printing. The multiresponsive AzPy moieties offer materials with photo- and chemical-controllable deformation [128]. We are optimistic about the future of AzPy-containing supramolecular materials and expect more breakthroughs in theory and practice in this field.

## Figures and Tables

**Figure 1 molecules-27-03977-f001:**
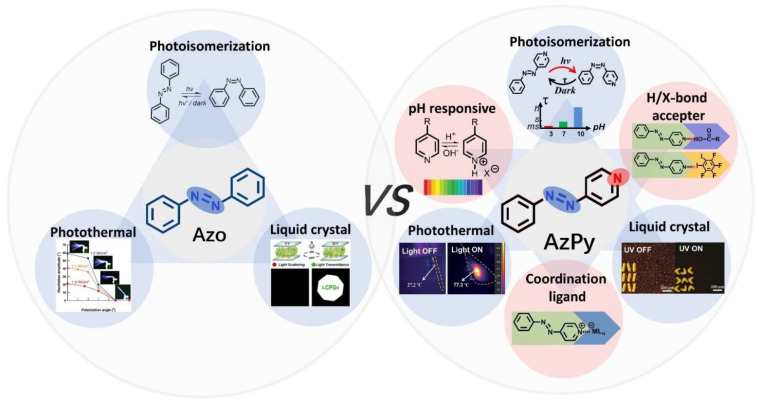
Scheme of multiresponsive properties of azobenzene and azopyridine (AzPy) and their applications from photonics to nanotechnology.

**Figure 2 molecules-27-03977-f002:**
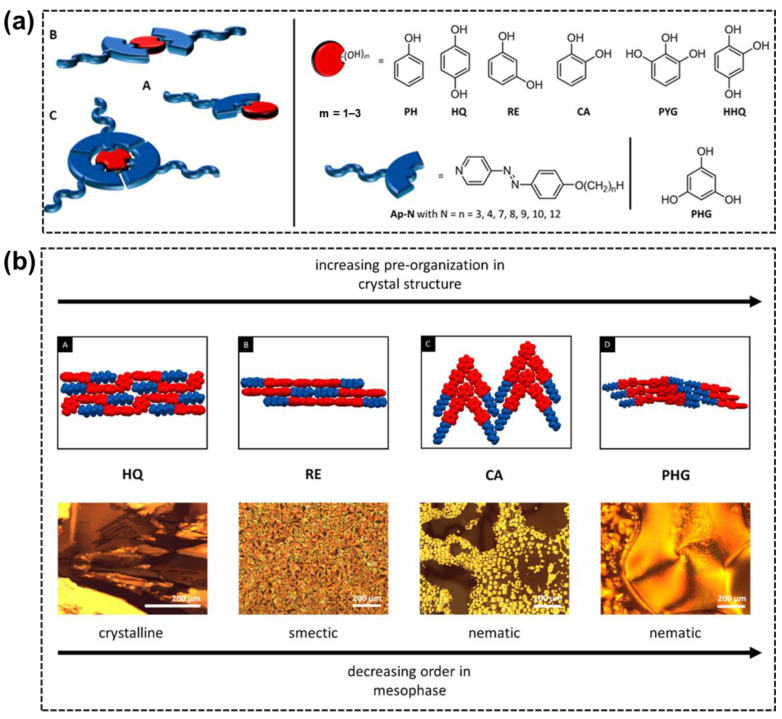
(**a**) Systematic study of hydrogen-bonded assemblies schematically drawn for 1:1 (A), 2:1 (B), and 3:1 (C) complexes. The used core units (red) phenol (PH), hydroquinone (HQ), resorcinol (RE), catechol (CA), pyrogallol (PYG), hydroxyhydroquinone (HHQ), and phloroglucinol (PHG) were mixed with azopyridine (blue) to form the HBAs. (**b**) Representative views of the intermolecular packing of HQ, RE, CA, and PHG assemblies (A–D, upper images), as observed with respect to the solid-state structures. The observations under cross-polarized microscope (lower images) show the correlation between the structural morphology and thermal behavior of the investigated hydrogen-bonded assemblies. Reprinted with permission from Reference [36]. Copyright 2017, American Chemical Society.

**Figure 3 molecules-27-03977-f003:**
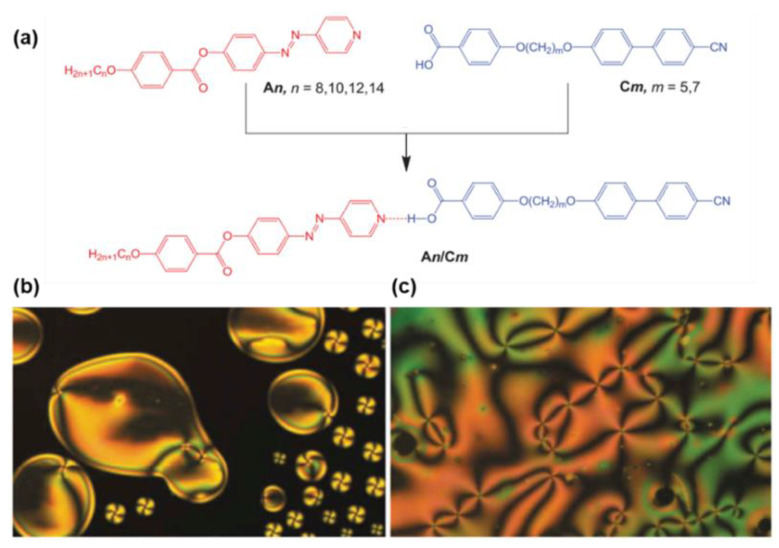
(**a**) Synthetic route to the hydrogen-bonded complex An/Cm; (**b**) Optical micrographs observed for the nematic phase of the supramolecular complex A8/C5 in homeotropic cell: nematic droplets at the transition from the isotropic liquid at T = 191 °C; (**c**) schlieren texture of the nematic phase showing four-brush defects at T = 150 °C. Reprinted with permission from Reference [33]. Copyright 2019, Taylor & Francis.

**Figure 4 molecules-27-03977-f004:**
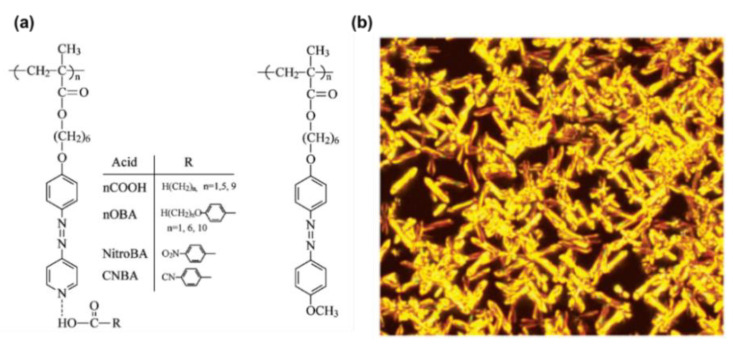
(**a**) Chemical structures of the amorphous azopyridine polymer and a liquid–crystalline azobenzene polymer counterpart. (**b**) Polarizing optical micrographs for PAzPy complexed with (**a**) acetic acid (1COOH), annealed at 63 °C. Reprinted with permission from Reference [52]. Copyright 2004, American Chemical Society.

**Figure 5 molecules-27-03977-f005:**
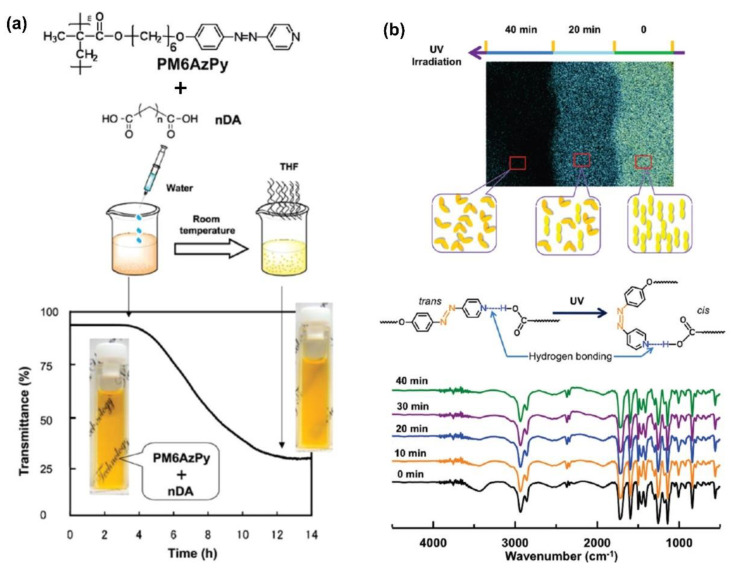
(**a**) Polymer structure and preparation of hydrogen−bonded liquid−crystal materials. (**b**) Photoresponse of the PM6AzPy−10DA film, top: POM image, bottom: FTIR spectra. Adapted with permission from [58]. Copyright 2011, American Chemical Society.

**Figure 6 molecules-27-03977-f006:**
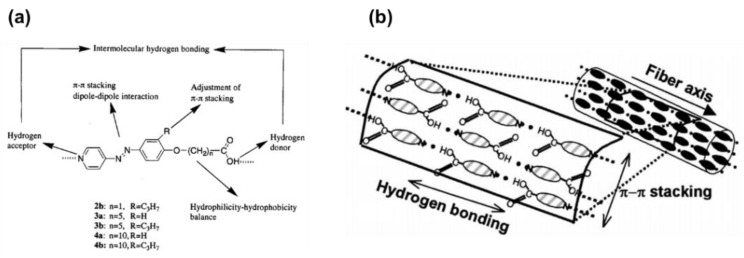
(**a**) Head-to-tail coupled and hydrogen-bonded azopyridine derivatives. (**b**) Schematic representation of molecular assemblages formed from azopyridine carboxylic acids. Reprinted with permission from [60]. Copyright 2011, American Chemical Society.

**Figure 7 molecules-27-03977-f007:**
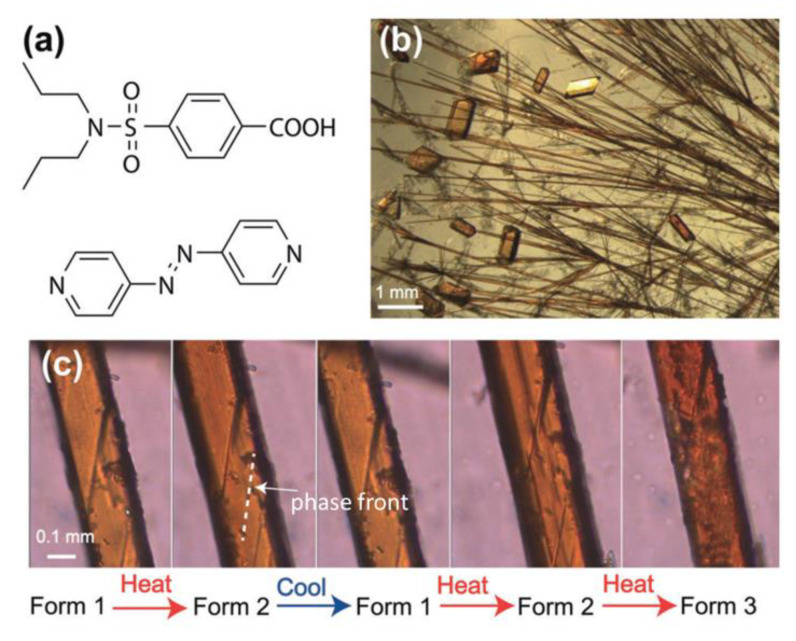
(**a**) Molecular structures of probenecid and 4,4′-azopyridine; (**b**) Optical microscopic image showing the crystal habits of the two polymorphs of the co-crystal, form 1 (acicular crystals) and form 3 (blocky crystals); (**c**) Thermal microscopy images showing the progression of the habit plane (phase front) during the reversible phase transition from form 1 to form 2 and during the irreversible phase transition from form 2 to form 3. Reprinted with permission from [66]. Copyright 2018, John Wiley & Sons.

**Figure 8 molecules-27-03977-f008:**
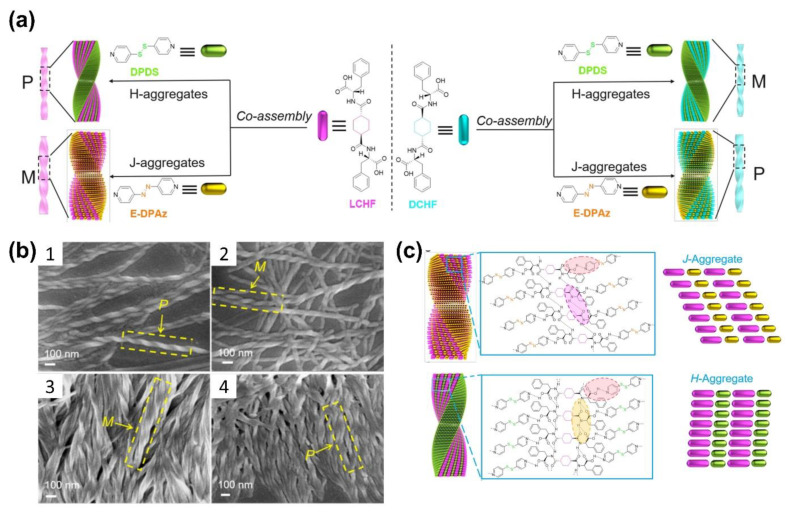
(**a**) Schematic presentation of the helical inversion triggered by H-aggregation of DPDS and J-aggregation of EDPAz in the co-assembly process with (left) LCHF or (right) DCHF. M represents left-handed helical nanofibers, and P represents right-handed helical nanofibers; (**b**) SEM images of (1) LCHF + DPDS with P-helical nanofibers, (2) DCHF + DPDS with M-helical nanofibers, (3) LCHF + EDPAz with M-helical nanofibers, and (4) LCHF + EDPAz with P-helical nanofibers; (**c**) Schematic representation for loose J-type packing of LCHF + EDPAz-induced left-handed co-assembly (top) and compact H-type stacking of LCHF + DPDS-induced right-handed co-assembly (bottom). Reprinted with permission from [64]. Copyright 2018, American Chemical Society.

**Figure 9 molecules-27-03977-f009:**
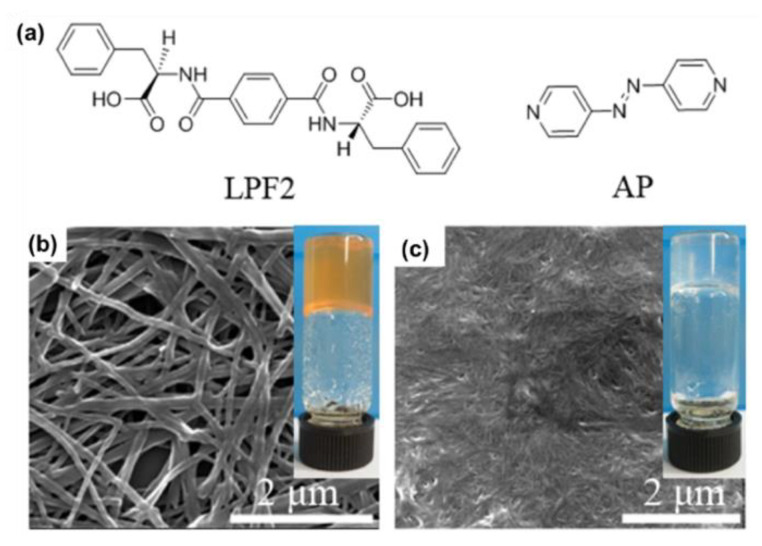
(**a**) Chemical structures of the gelator LPF2 and azobenzene derivative AP; SEM images of (**b**) LPF2-AP gel and (**c**) LPF2 gel. (Inset) Photographs of (**b**) LPF2-AP hydrogel and (**c**) LPF2 hydrogel. Reprinted with permission from [65]. Copyright 2015, American Chemical Society.

**Figure 10 molecules-27-03977-f010:**
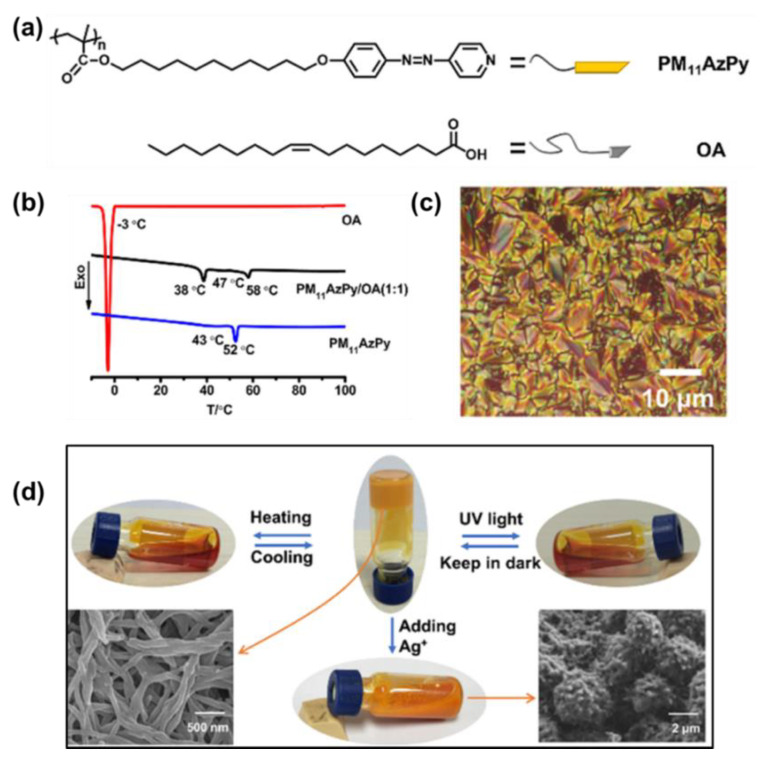
(**a**) Chemical structures of the azopyridine-containing polymer (PM_11_AzPy) and oleic acid (OA); (**b**) DSC curves (second cooling scan) of OA, PM_11_AzPy, and PM_11_AzPy−OA (1:1); (**c**) One POM image of the compound PM_11_AzPy−OA (1:1) annealed at 46 °C, scale bar: 10 μm; (**d**) Photos of the organogel in response to heating, UV light, and adding Ag^+^ (inset SEM image: PM_11_AzPy−OA (1:20) xerogel and precipitates from the organogel after adding Ag^+^). Reprinted with permission from [68]. Copyright 2019, American Chemical Society.

**Figure 11 molecules-27-03977-f011:**
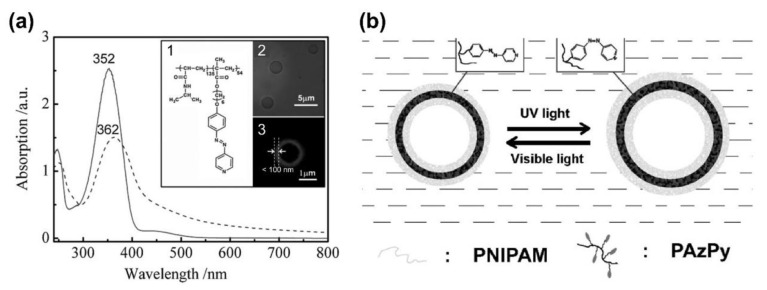
(**a**) Absorption spectra of the copolymer in THF solution (solid line) and in the mixture of THF and H_2_O as vesicles (dashed line). The inset shows (1) the chemical structure of the deblock copolymer; (2) images were obtained using the optical microscope and (3) laser scanning confocal microscopy, respectively; (**b**) Schematic representation of UV-induced swelling and shrinking behaviors from block copolymer vesicles. Reprinted with permission from [69]. Copyright 2008, WILEY.

**Figure 12 molecules-27-03977-f012:**
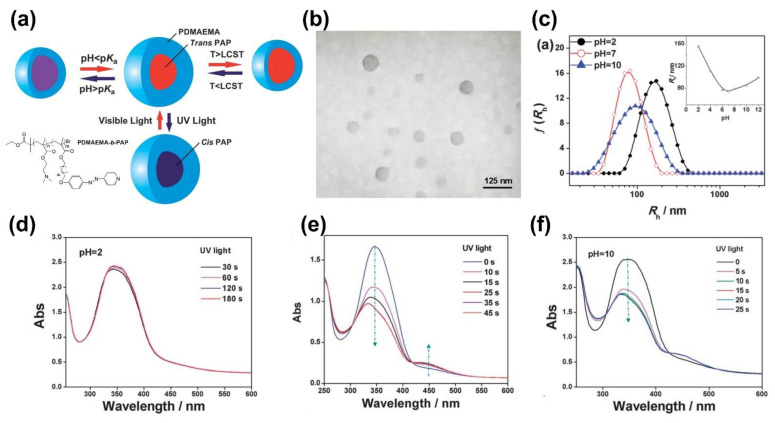
(**a**) Schematic representation of triple responses from the shell to the core of PDMAEMA-*b*-PAzPy micelles under temperature, pH, and light stimuli; (**b**) TEM image of PDMAEMA-*b*-PAzPy micelles (25 °C and pH 7); (**c**) Z-averaged size distributions of PDMAEMA-*b*-PAzPy micelles at different pH values (inset: pH-dependence of hydrodynamic radius). UV/Vis absorption spectra of PDMAEMA-*b*-PAP micelles solution at (**d**) pH 2, (**e**) pH 7, and (**f**) pH 10 upon irradiation at 365 nm for different times (concentration: 0.15 mg mL^−1^). Reprinted with permission from [72]. Copyright 2013, Royal Society of Chemistry.

**Figure 13 molecules-27-03977-f013:**
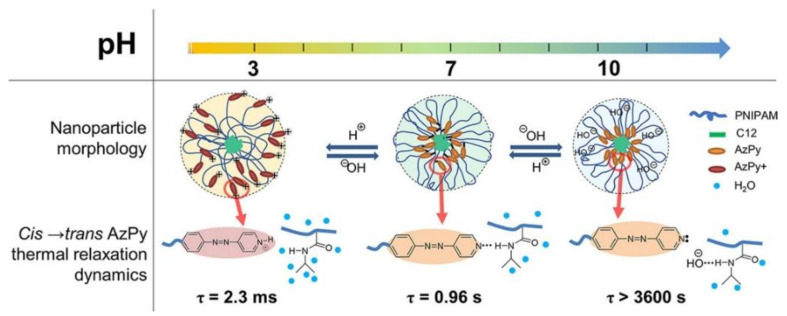
Schematic representation of C12-PN-AzPy nanoparticles dispersed in water of pH 3, 7, and 10 based on data from LS, FTIR, and ^1^H NMR measurements and on the kinetics of the *cis*-to-*trans* thermal relaxation of azopyridine. Reprinted with permission from [24]. Copyright 2019, American Chemical Society.

**Figure 14 molecules-27-03977-f014:**
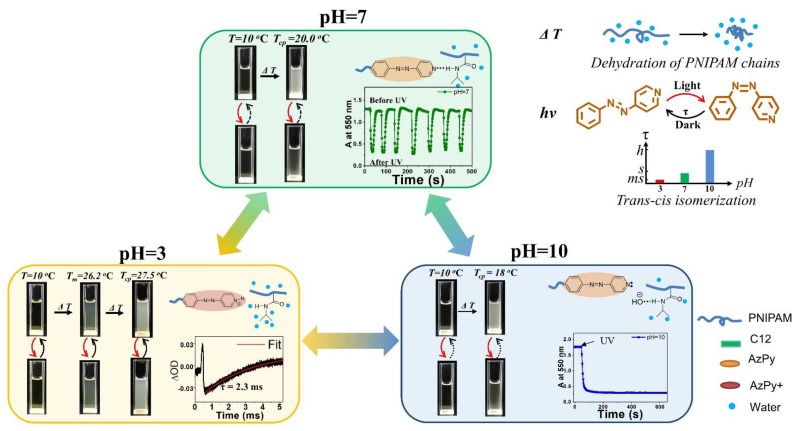
pH− and UV−induced thermoresponsive properties of C12−PN−AzPy (the temperatures correspond to a polymer of M_n_~7000 g mol^−1^, ΔOD: difference in the optical density before and after irradiation). Reprinted with permission from [75,79]. Copyright 2019 and 2020, Royal Society of Chemistry.

**Figure 15 molecules-27-03977-f015:**
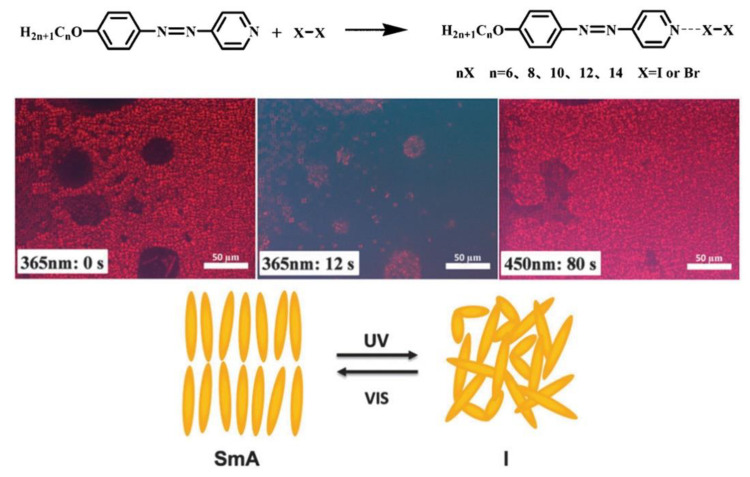
Possible molecular scheme of halogen complexes of AnAzPy and POM observation of 12I at its LC phase upon UV irradiation. The right picture was obtained after irradiation with visible light for 80 s. Reprinted with permission from [89]. Copyright 2014, Royal Society of Chemistry.

**Figure 16 molecules-27-03977-f016:**
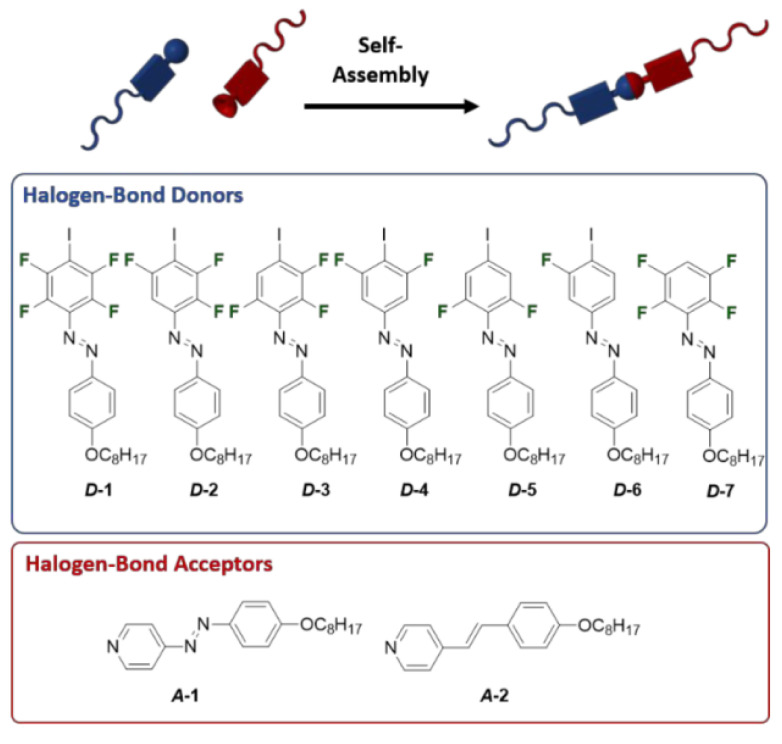
Modular approach to investigate the impact of the strength of the X-bond on the liquid–crystalline behavior of the supramolecular assemblies. Reprinted with permission from [92]. Copyright 2019, American Chemical Society.

**Figure 17 molecules-27-03977-f017:**
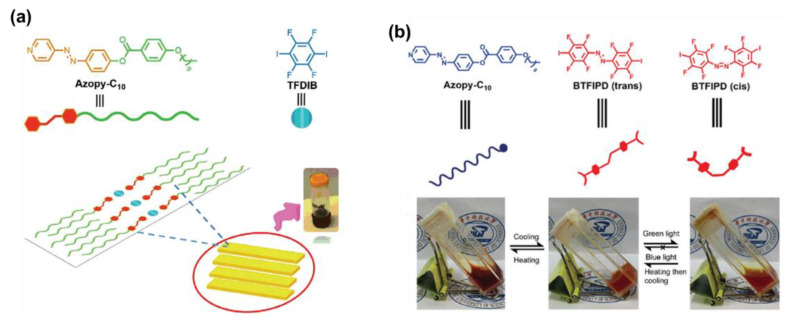
Illustrative strategy for preparing the supramolecular gel driven by X-bond. (**a**) Gel prepared from AzPy-C10 and 1,4-tetrafluorodiiodobenzene(TFDIB); Reprinted with permission from [93]. Copyright 2019 Wiley; and (**b**) gel prepared from AzPy-Cn (*n* = 8, 10, and 12) and 1,2-bis(2,3,5,6-tetrafluoro-4-iodophenyl)diazene (BTFIPD). Reprinted with permission from [94]. Copyright 2019, Royal Society of Chemistry.

**Figure 18 molecules-27-03977-f018:**
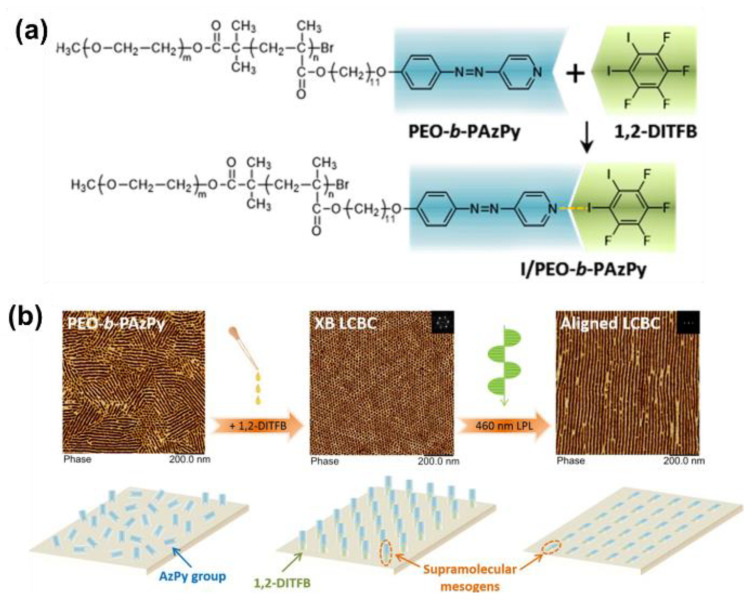
(**a**) Chemical structures of the 1:1 X-bond complex between PEO-*b*-PAzPy and 1,2-diiodo-3,4,5,6-tetrafluorobenzene (1,2-DITFB). (**b**) Photo-reorientation of supramolecular mesogens and ordered microphase separation nanostructures in the XB-involved supramolecular LCBC film. Reprinted with permission from [95]. Copyright 2020, American Chemical Society.

**Figure 19 molecules-27-03977-f019:**
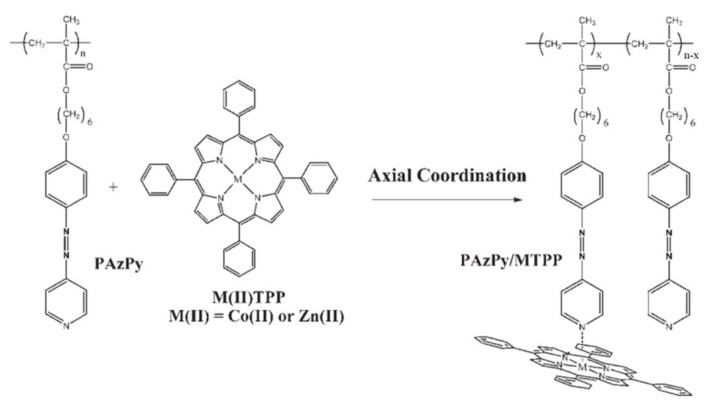
Complexation via axial coordination between PAzPy and ZnTPP or CoTPP. Reprinted with permission from [106]. Copyright 2006, Wiley.

**Figure 20 molecules-27-03977-f020:**
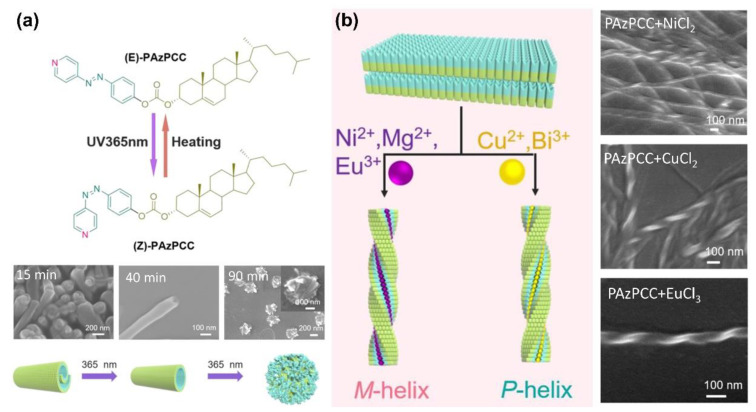
(**a**) Chemical structure and photoinduced dimensional transition of the PAzPCC self-assembly, insets SEM images of assemblies upon the irradiation of 365 nm light for different times. (**b**) The coordination interaction of AzPy gelator with Ni^2+^, Eu^3+^, Cu ^2+^, and Bi^3+^ produce helical chirality nanofibers. Reprinted with permission from [109]. Copyright 2018, American Chemical Society.

**Figure 21 molecules-27-03977-f021:**
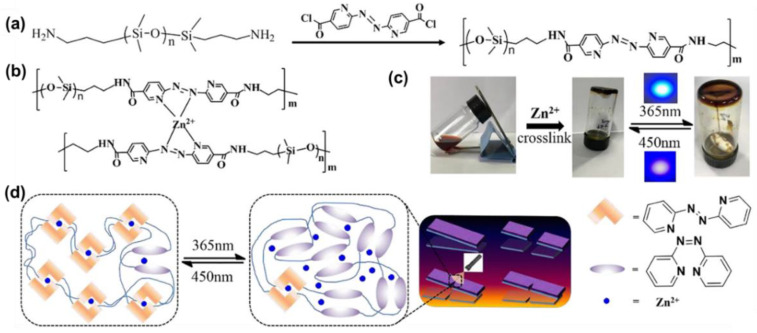
(**a**) The synthetic route of AzPy-PDMS; (**b**) the structure of Zn(AzPy)2-PDMS and (**c**) photographs of the gelation formation of an AzPy -PDMS toluene solution (70 mg/mL, 3 mL) upon the addition of methanol solution of Zn(OTf)_2_ (0.18 M, 100 μL); (**d**) The schematic diagram of the light-healing process. Reprinted with permission from [110]. Copyright 2020, Elsevier.

**Figure 22 molecules-27-03977-f022:**
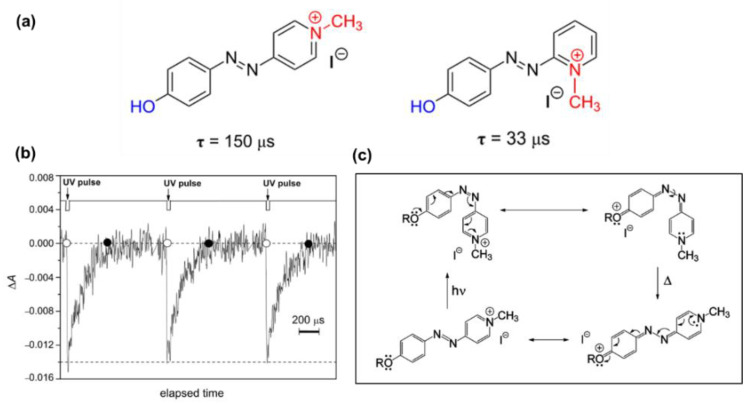
(**a**) Effect of the presence of a positively charged nitrogen as an electron−withdrawing group on the thermal relaxation time at 298 K, τ, for the type−I azoderivative. (**b**) Oscillation of the optical density of an ethanol solution of azo-dye generated by UV-light irradiation. (**c**) Mechanism proposed for the thermal *cis*-to-*trans* isomerization process for the azopyridinium methyl iodide salt. Reprinted with permission from [73,114]. Copyright 2010, American Chemical Society and 2011, Royal Society of Chemistry.

**Figure 23 molecules-27-03977-f023:**
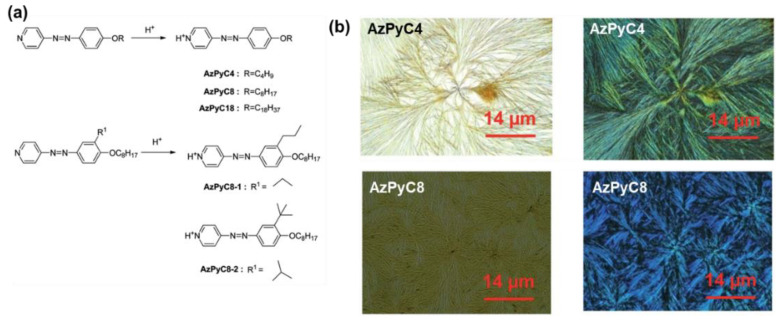
(**a**) Structure and synthesis of low-molecular-weight compounds for fabrication of organic nanofibers. (**b**) Optical images and POM images of AzPyC4 and AzPyC8. Reprinted with permission from [121]. Copyright 2015, Royal Society of Chemistry.

**Figure 24 molecules-27-03977-f024:**
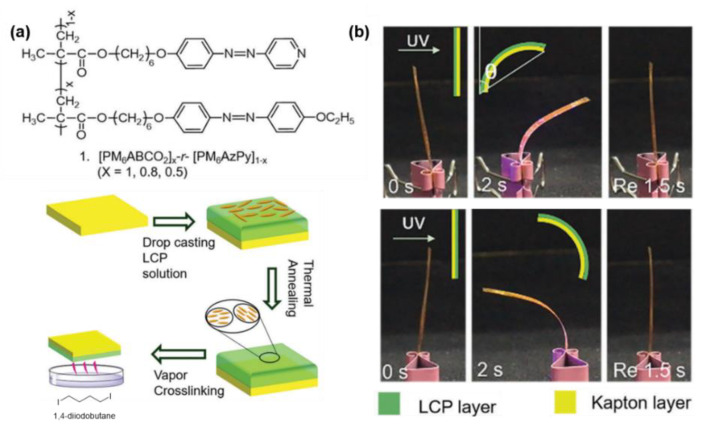
(**a**) Structure of azopyridine containing polymer crosslinked by quaternization reaction of pyridine. (**b**) Photomechanical behaviors of bimorph composites. Reprinted with permission from [123]. Copyright 2019, Royal Society of Chemistry.

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
