# Peer review of "Recent Progress in Azopyridine-Containing Supramolecular Assembly: From Photoresponsive Liquid Crystals to Light-Driven Devices"

_molecules, 2022, doi:10.3390/molecules27133977_

Round 1

Reviewer 1 Report

The paper satisfies the requirements demanded for a review. It presents a detailed discussion of a clearly defined subject, it contains a lot of information and citations for the interested readers. The emphasis is placed on the light-induced isomerization of azo-group based compounds, which is a hot topic with many potential applications. I recommend the review for publication is its present form.

Author Response

REVIEWER REPORT(S):

Referee: 1

Comments to the Author

The paper satisfies the requirements demanded for a review. It presents a detailed discussion of a clearly defined subject, it contains a lot of information and citations for the interested readers. The emphasis is placed on the light-induced isomerization of azo-group based compounds, which is a hot topic with many potential applications. I recommend the review for publication is its present form.

RE: We highly appreciated the positive evaluation, and we hope this review paper can be helpful for interested readers in fields of light-driven materials.

Reviewer 2 Report

The idea underlying the review of Ren and co-authors is very good. The authors have significant experience in this field. Although so much has already been told about photoswitchability of azobenzene-based supramolecular assemblies and azopyridine- and phenylazopyridine-based molecular switches have been reviewed by Crespi et al. (Nat Rev Chem 2019, 3, 133–146), the topic can be interesting for researchers working in the field of liquid crystals, gels, polymers etc. However, some improvements are needed. The present review can be accepted for publication after considering the following:

i) I am well aware that the authors are not English native speakers, but a serious language editing is needed. Even in the very first sentence: Light has incomparable advantages for it is clean, fast and precise character, which is possible to controlled through specific time and space scales. A native speaker should be consulted during the revision process.

ii) The precision is lacking here and there, e.g. Very recently, several new types of AzPy-containing supramolecular liquid crystal-line with liner-shape, [34,35] rod-like shape, [30] chair/V-shape, [36,37] taper-shape [38] as well as a core-fluorination substituent at different positions were reported. [38,39] (p. 4, line 129; firstly, linear-shape and rod-like shape are the synonyms, then the substitution pattern is at the end of the sentence describing the different geometries…)

iii) The directionality of non-covalent interactions as HB or XB has been used to build different supramolecular assemblies involving a pyridine unit as the accepting one. The analogy between HB and XB underpinning the approaches in the parts 2 and 3 should be presented in a clearer manner. Furthermore, the authors present the theoretical background that can be easily found elsewhere, while the selection of the examples involving azopyridine is rather modest. Please include more examples of Giese, Alaasar etc.

iv) Do the authors need the copyright agreement to present the Figures originally published in other papers?

Author Response

Referee: 2

Comments to the Author

The idea underlying the review of Ren and co-authors is very good. The authors have significant experience in this field. Although so much has already been told about photoswitchability of azobenzene-based supramolecular assemblies and azopyridine- and phenylazopyridine-based molecular switches have been reviewed by Crespi et al. (Nat Rev Chem 2019, 3, 133–146), the topic can be interesting for researchers working in the field of liquid crystals, gels, polymers etc. However, some improvements are needed. The present review can be accepted for publication after considering the following:

  1. i) I am well aware that the authors are not English native speakers, but a serious language editing is needed. Even in the very first sentence: Light has incomparable advantages for it is clean, fast and precise character, which is possible to controlled through specific time and space scales. A native speaker should be consulted during the revision process.

RE: Thanks for the referees' valuable comments. It points out our English writing may cause some misunderstandings to the possible readers. We have carefully checked and improved the English writing in the revised manuscript. All that revised was marked in blue (see the revised version-labeled).

Light is absolutely charming as it is an abundant and clean energy, which has been widely utilized to manipulate photoresponsive materials remotely, precisely and instantly. In the past few decades, the development of photoresponsive materials has attracted extensive attention in view of their wide-ranging applications, including nanotechnology, light-driven actuators, [1,2] drug delivery systems, [3] controlled biological systems, [4,5] and many more. [6-9]

  1. ii) The precision is lacking here and there, e.g. Very recently, several new types of AzPy-containing supramolecular liquid crystal-line with liner-shape, [34,35] rod-like shape, [30] chair/V-shape, [36,37] taper-shape [38] as well as a core-fluorination substituent at different positions were reported. [38,39] (p. 4, line 129; firstly, linear-shape and rod-like shape are the synonyms, then the substitution pattern is at the end of the sentence describing the different geometries…)

RE: Thanks for the kindly reminding of the referee. We are sorry for the ambiguous expression in the submitted version. The discussion was re-written based on the comments in the revised version.

Very recently, several new types of AzPy-containing supramolecular liquid crystals with different geometries were reported, e.g. rod-like shape, [33,45,46] chair/V-shape, [47,48] taper-shape [49] as well as bent-shape. [49,50]

iii) The directionality of non-covalent interactions as HB or XB has been used to build different supramolecular assemblies involving a pyridine unit as the accepting one. The analogy between HB and XB underpinning the approaches in the parts 2 and 3 should be presented in a clearer manner. Furthermore, the authors present the theoretical background that can be easily found elsewhere, while the selection of the examples involving azopyridine is rather modest. Please include more examples of Giese, Alaasar etc.

RE: This comment is extremely important. It points to a problem in the way we introduced the hydrogen bond (HB) and halogen bond (XB) in submitted version. The reviewer, as most readers of molecules, is certainly familiar with the basic concepts of non-covalent interactions of HB and XB. However, we did not provide detailed information (bond strength, direction, and geometry). The submitted version did not discuss the analogy between these two similar interactions. Thanks again for the constructive comments. In the revised version, we provided the missing information accordingly.

The lone-pair electron in AzPy enables it to interact with various hydrogen bond (H-bond) donors easily, acting as building block for supramolecular assembly via H-bonding. The strength of H-bond is around 2-160 kJ/mol based on its length and geometry.[25] The shorter the length and the closer B-H···A angle to 180 °, the stronger the H-bond, and vice versa.[26,27] The most widely investigated H-bond donor is carboxylic acid derivatives, and the equilibrium constant (Ka) of the pyridyl/carboxylic acid complex was estimated to be five times stronger than that of the carboxylic acid dimer. [28] Because of its strength and directionality, H-bonds usually act as a powerful tool to guide supramolecular assembly in nature (e.g., proteins, DNA). In addition, a significant number of assemblies such as liquid crystals (LCs), fibers, films, and gels have been extensively studied. As shown below, we summarize the recent progress of AzPy-based supramolecules and the self-assembled structures via H-bond.

Similar to H-bond, the AzPy group, as an electron-rich Lewis base, can act as a halogen-bond (X-bond) acceptor due to the lone pair of electrons of a nitrogen atom in the pyridine ring. The nitrogen atom interacts non-covalently with an electropositive Lewis acid (H-bond donor), usually σ-hole of a halogen atom, to form X-bond.[81,82] The X-bond strength ranges from 10 kJ/mol to 150 kJ/mol depending on the interaction partners, which is slightly weaker than the corresponding H-bond and can be modulated by the compositional of donor atoms as well as its substitution groups. Typically, X-bond strength has a general order of I > Br > Cl > F, where a fluorine atom can only be an X-bond donor when attached to strong electron-withdrawing groups.[83] In addition, X-bonding is more directional than H-bond because the σ-hole of a halogen atom is narrowly confined on the elongation of the R-X covalent bond axis. Hence the B-X···A angle is closer to 180° than H-bond. [84,85]

Benefiting from the strength and more directional nature of X-bond, supramolecular chemists can easily design and prepare diverse smart materials.

For the selected examples, we are sorry that several important articles in this area was missed. In the revised version, azopyridine-based liquid crystalline works from Giese and Alaasar’s group were cited.

In their following works, hierarchical supramolecular liquid crystals were obtained by self-assembling different core units through H-bonding interaction. In these structures, AzPy derivatives act as H-bond acceptors and aromatic polyols or polyphenols act as H-bond donors. For example, phloroglucinol [37] resveratrol,[38] oxyresveratrol, butein, isoliquiritigenin, piceatannol,[39] polycatenars [40] and other ortho-substituted phloroglucinol (e.g. 2-fluoro,[41,42] cyan, nitro [43]). The relationship between the core structure and light-responsive liquid crystal properties was systematically investigated. Combined with detailed computational analysis with temperature-dependent FTIR results, they revealed an entropy-driven unfolding mechanism of the assembly.[44] Several H-bond liquid crystals exhibit rapid photo-response and the broad-range blue phase, showing potential applications in organic optoelectronics as sensors or optical gates.

  1. Pfletscher, M.; Wölper, C.; Gutmann, J.S.; Mezger, M.; Giese, M. A modular approach towards functional supramolecular aggregates – subtle structural differences inducing liquid crystallinity. Chem. Commun. 2016, 52, 8549-8552.
  2. Blanke, M.; Balszuweit, J.; Saccone, M.; Wolper, C.; Doblas, J.D.; Mezger, M.; Voskuhl, J.; Giese, M. Photo-switching and -cyclisation of hydrogen bonded liquid crystals based on resveratrol. Chem. Commun. 2020, 56, 1105-1108.
  3. Balszuweit, J.; Blanke, M.; Saccone, M.; Mezger, M.; Daniliuc, C.G.; Wölper, C.; Giese, M.; Voskuhl, J. Naturally occurring polyphenols as building blocks for supramolecular liquid crystals – substitution pattern dominates mesomorphism. Mol. Syst. Des. Eng. 2021, 6, 390-397.
  4. Alaasar, M.; Cai, X.; Kraus, F.; Giese, M.; Liu, F.; Tschierske, C. Controlling ambidextrous mirror symmetry breaking in photosensitive supramolecular polycatenars by alkyl-chain engineering. J. Mol. Liq. 2022, 351, 118597.
  5. Malotke, F.; Saccone, M.; Wölper, C.; Dong, R.Y.; Michal, C.A.; Giese, M. Chiral mesophases of hydrogen-bonded liquid crystals. Mol. Syst. Des. Eng. 2020, 5, 1299-1306.
  6. Saccone, M.; Pfletscher, M.; Dautzenberg, E.; Dong, R.Y.; Michal, C.A.; Giese, M. Hydrogen-bonded liquid crystals with broad-range blue phases. J. Mater. Chem. C 2019, 7, 3150-3153.
  7. Saccone, M.; Pfletscher, M.; Kather, S.; Wölper, C.; Daniliuc, C.; Mezger, M.; Giese, M. Improving the mesomorphic behaviour of supramolecular liquid crystals by resonance-assisted hydrogen bonding. J. Mater. Chem. C 2019, 7, 8643-8648.
  8. Pfletscher, M.; Wysoglad, J.; Gutmann, J.S.; Giese, M. Polymorphism of hydrogen-bonded star mesogens – a combinatorial DFT-d and FTIR spectroscopy study. RSC Adv. 2019, 9, 8444-8453.

  1. iv) Do the authors need the copyright agreement to present the Figures originally published in other papers?

RE: Thanks for the reminder of the referee, permission for the reuse of the figures was added in the revised version. Since we did not get the permission of Figure 13 (Nature 2017, 546, 632) and Figure 25 (Acta Polym. Sin. 2018, 1175), they were deleted, and all figure numbers were renewed in the submitted version.

Reviewer 3 Report

Comments on the manuscript ID molecules-1743724 by Ren et al. titled “Recent progress in azopyridine-containing supramolecular assembly: from photoresponsive liquid crystal to light-driven devices”:

The authors reviewed the progress in azopyridine (AzPy) supramolecular assembles. They have experience in the field and have published several papers on the topic (references 24, 69). Ren, Yang and Winnik have recently published a similar review titled “Azopyridine: a smart photo- and chemo-responsive substituent for polymers and supramolecular assemblies” (DOI: 10.1039/D0PY01093F (Minireview) Polym. Chem., 2020, 11, 5955-5961). Figure 15 was published with permission in ref. 69, but also in this minireview (Figure 4, page 5959, Reprinted with permission from ref. 47 Copyright (2019) Royal Society of Chemistry).  The minireview and the materials that have been already published should be cited correctly in the current review.

The review does not cover theoretical studies of AzPy systems. The computational methods are powerful tool for design of single chromophores and supramolecular assembles. The authors noted that “is usually necessary to adjust its photochemical property from the molecular level”, but these type of predictions can be done easily and reliably by quantum-chemical calculations. The authors should also review these studies.

Minor points:

1)      Abstract: the authors should not use “azo” instead of azo group, azo dyes: “Azo is one of the most famous photochromophore in the past few decades for its reversible archetypical molecular switches upon photo irradiation. To meet ever-increasing requirements for applications in materials science, biomedicine and light-driven devices, it is usually necessary to adjust its photochemical property from the molecular level by adding substituents on the benzene rings of azo.”;

2)      Reference formatting.

Author Response

Referee: 3

Comments on the manuscript ID molecules-1743724 by Ren et al. titled “Recent progress in azopyridine-containing supramolecular assembly: from photoresponsive liquid crystal to light-driven devices”:

  1. i) The authors reviewed the progress in azopyridine (AzPy) supramolecular assembles. They have experience in the field and have published several papers on the topic (references 24, 69). Ren, Yang and Winnik have recently published a similar review titled “Azopyridine: a smart photo- and chemo-responsive substituent for polymers and supramolecular assemblies” (DOI: 10.1039/D0PY01093F (Minireview) Polym. Chem., 2020, 11, 5955-5961). Figure 15 was published with permission in ref. 69, but also in this minireview (Figure 4, page 5959, Reprinted with permission from ref. 47 Copyright (2019) Royal Society of Chemistry). The minireview and the materials that have been already published should be cited correctly in the current review.

RE: Thanks for the kindly reminding from the referee. Our recent minireview paper (Polym. Chem., 2020, 11, 5955-5961) mainly discussed the azopyridine derivatives in aqueous media with H-bond controlled cis-to-trans relaxation property. And the current review summarized the recent accomplishments in azopyridine containing supramolecular, including liquid crystals, gels, and films. We mentioned several exciting examples, and hopefully, this review paper may be a benefit for possible readers. Finally, the references of Figure 14 (Figure 15 in the submitted version) were renewed according to the referee’s comments.

Figure 14. pH- and UV-induced thermoresponsive properties of C12-PN-AzPy (the temperatures correspond to a polymer of Mn ∼ 7000 g mol-1, ΔOD: difference in the optical density before and after irradiation) Reprinted with permission from [74] and [78]. Copyright 2019 and 2020 Royal Society of Chemistry.

  1. ii) The review does not cover theoretical studies of AzPy systems. The computational methods are powerful tool for design of single chromophores and supramolecular assembles. The authors noted that “is usually necessary to adjust its photochemical property from the molecular level”, but these type of predictions can be done easily and reliably by quantum-chemical calculations. The authors should also review these studies.

RE: This comment is extremely important and constructive. We believe that computer simulation can provide important theoretical support for the design of supramolecular assemblies. Currently, the simulation methods usually used in light-responsive materials can be mainly divided into the quantum simulation, molecular dynamic simulation (MD), and finite element simulation (FEM) from different scales. There are constantly emerging modified simulation methods in recent years and have been reviewed in many articles (Chem. Soc. Rev., 2012, 41, 1809; Phys. Sci. Rev. 2017; 20170138). Hence, in the revised version, we added several discussions about computational research on azopyridine-containing liquid crystals and supramolecular.

Very recently, several new types of AzPy-containing supramolecular liquid crystals with different geometries were reported, e.g. rod-like shape, [33,45,46] chair/V-shape, [47,48] taper-shape [49] as well as bent-shape. [49,50] It was suggested that the chair-shaped conformers were more stable than V-shaped isomeric complexes. [47] In addition, the rod-like conformation exhibited only enantiotropic nematic phase over a broad range of temperature, regardless of the terminal alkyl chain lengths at the pyridine-based component or the length of the flexible spacer on the benzoic acid derivatives (Figure 3). [33] The molecular conformation and the thermal parameters of the complexes were also confirmed by theoretical calculations via density functional theory (DFT). Ahipa and co-workers reviewed the recent progress of heterodimeric H-bonded mesogens containing the pyridyl moiety. [51]

Wei and coworkers designed a 2:1 cocrystal of (E)-4-((4-(propyloxy)phenyl)diazenyl)pyridine (APO3C) and tetrafluoroterephthalic acid (TFTA), the result cocrystal exhibit photoinduced rotation under UV light. With the help of molecular simulation, the crystal habits and intermolecular interactions within these two assemblies were clarified. The pointer-like photomechanical rotation can be attributed to the opposing force between the rotating of crystals caused by trans-cis isomerization of APO3C molecules and the limited action of diagonal arrangement. [67]

In addition to experimental achievements, theoretical calculations have been regarded as a powerful tool for design of liquid crystals and supramolecular materials. For example, Jimmy and coworkers recently predicted the dynamic photoreaction pathways of azobenzene with the help of first-principles simulations of nonadiabatic dynamics following excitation to both the π-π* and the n-π* states. They indicated two distinct S1 decay pathways, the reactive twisting and the unreactive planar pathway. Moreover, the unreactive pathway upon π-π* excitation largely accounts for the wavelength-dependent behaviour of azobenzene. [80] Karolina and coworkers utilized DFT to optimize ground and transition state geometries, the density of states, and electronic structures of azobenzene structures, indicating that the possible mechanism of the cis-trans isomerization process for 4-(4-hydroxyphenylazo)pyridine has a different relaxation time than other azo-compounds. [13]

With the help of DFT calculation, molecular electrostatic potentials of iodine atom increase (more positive) as the number of fluorine atoms increases, demonstrating a simplistic“the shorter, the stronger” view of H-bonding interaction.

iii) Minor points:

  • Abstract: the authors should not use “azo” instead of azo group, azo dyes: “Azo is one of the most famous photochromophore in the past few decades for its reversible archetypical molecular switches upon photo irradiation. To meet ever-increasing requirements for applications in materials science, biomedicine and light-driven devices, it is usually necessary to adjust its photochemical property from the molecular level by adding substituents on the benzene rings of azo.”;

RE: The text is revised accordingly. It now reads:

Azobenzene derivatives are one of the most famous photoresponsive chromophores in the past few decades for their reversible molecular switches upon irradiation of actinic light. To meet ever-increasing requirements for applications in materials science, biomedicine and light-driven devices, it is usually necessary to adjust its photochemical property from the molecular level by changing substituents on the benzene rings of azobenzene groups.

  • Reference formatting.

RE: All reference format is carefully checked and revised accordingly.

Round 2

Reviewer 3 Report

The authors have addressed all my comments.